# The Observed Impact of the South Asian Summer Monsoon on Land-Atmosphere Heat Transfers and Its Inhomogeneity over the Tibetan Plateau

Hongyi Li [1], Libo Zhou [2] and Ge Wang [3,*]

1   China Meteorological Administration Training Centre, Beijing 100081, China; lihongyi@cma.gov.cn
2   State Key Laboratory of Atmospheric Boundary Layer Physics and Atmospheric Chemistry & Department of Lower Atmosphere Observation and Research, Institute of Atmospheric Physics, Chinese Academy of Sciences, Beijing 100029, China; zhoulibo@mail.iap.ac.cn
3   Institute of Plateau Meteorology, China Meteorological Administration, Chengdu 610072, China
*   Correspondence: wg800110@aliyun.com

**Abstract:** To promote Tibetan meteorological research, the third Tibetan Plateau (TP) Experiment for atmospheric sciences (TIPEX III) has been carried out over the plateau region since 2014, with near-surface heat fluxes measured at different sites. Using the observational data of near-surface heat fluxes measured at 8 plateau stations in TIPEX III, as well as the ECMWF ERA Interim reanalysis data, the land-atmosphere heat transfers over different regions of TP and their responses to the South Asian summer monsoon (SASM) during active/break periods were investigated. Inhomogeneity was found in the land-atmosphere heat transfers over the plateau, with large differences among plateau stations. During the observation period, the daily averaged total heat transfer (the sum of sensible and latent heat flux) varied from 70.2 to 101.2 Wm$^{-2}$ among the 8 plateau stations, with the sensible heat flux from 18.8 to 60.1 Wm$^{-2}$ and the latent heat flux from 10.1 to 74.7 Wm$^{-2}$. These heat transfers were strongly affected by the SASM evolution, but with strong inhomogeneities over the plateau stations. Overall, the more southern station locations exhibited more SASM impacts. The land-atmosphere heat transfers (the total, sensible and latent heat fluxes) were greatly weakened/strengthened during the SASM active/break period at Namco (southeast plateau), Baingoin (central plateau), Lhari (central plateau), and Nagqu (central plateau), which were closely related to the weakened/strengthened radiation conditions. However, the SASM impacts were quite small or even negligible for the other plateau stations, which complicated our conclusions, and further investigations are still needed.

**Keywords:** Tibetan Plateau; South Asian summer monsoon; land-atmosphere heat transfer; inhomogeneity

## 1. Introduction

As the "Third Pole of the Earth" and the "atmospheric water tower", the Tibetan Plateau (TP) has an area of about 2.5 million square kilometers and an average altitude greater than 4000 m. The strong solar radiation on this elevated surface makes the plateau a heating source for the atmosphere in the middle troposphere. This surface heating not only impacts the local weather and synoptic situations over the plateau, but also affects the climate and environment of East Asia and the Northern Hemisphere [1–4]. For example, Yanai et al. revealed that Tibetan surface heating has a great influence on the onset of the South Asian summer monsoon [2], and Zhou et al. noted that the possible impacts of the plateau surface heating could be expanded to the Northern Atlantic Ocean [4]. Zou showed the role of surface heating in the formation of the Tibetan ozone low [5].

To understand the surface heating over the Tibetan Plateau, many scientific experiments have been carried out, in particular, the First and Second Tibetan Plateau Atmospheric Scientific Experiments. Using the observational data, Ye and Gao studied the

land-air heat exchange and noted that sensible heat transfer dominates the surface heating over the plateau, especially in summer [1]. Gao et al., Bian et al., Li et al. and Zou et al. observed the land-air heat exchange at Nagqu (Central Tibet), Qamdo (East Tibet), Gerze (West Tibet), and the northern slope of Mt. Everest (South Tibet), respectively, in spring and early summer [6–9]. They showed that the total turbulent heat fluxes (defined as the sum of sensible and latent heat flux) over East, West, and Central Tibet were in the range of 80.0–144.0 Wm$^{-2}$, with a higher sensible heat transfer in the range of 43.0–86.0 Wm$^{-2}$, and a lower latent heat transfer with the range of 28.0–59.0 Wm$^{-2}$. However, Zou et al. found that the near-surface heat transfer in the Southeast Tibet in early summer was significantly different from that in the other Tibetan regions [10], with a total heat flux of 86.3 Wm$^{-2}$, a sensible heat flux of 22.9 Wm$^{-2}$ and a latent heat flux of 63.4 Wm$^{-2}$. The latent heat transfer dominates the land-air heat exchange in Southeast Tibet.

The South Asian summer monsoon (SASM) is an important component of the Asian monsoon, which has a great influence on the atmospheric processes over Asia [11–15]. The SASM usually starts in late May or early June, which is characterized by the formation of cyclonic vortices in the Bay of Bengal or in the Southeast Arabian Sea [11,16,17]. After its onset, the SASM develops during the summer and autumn, with several active and break periods observed, which are characterized by the heavy and light rainfalls associated with the different monsoon troughs over South Asia [12,18–20]. The SASM decays in late September or early October. The SASM mainly affects the Indian Peninsula and Indo-China Peninsula, and the affected areas could extend northwards to the Qinghai-Tibet Plateau and Southwest China [13,16]. Gao et al. showed that the precipitation over Southeast Tibet can be affected by the monsoon [21]. Zhou et al. [17,22–25] and Li et al. [26] found that the local atmospheric properties in the Himalayas and Southeast Tibet region are closely related to the SASM evolutions. Most recently, Zou et al. [9,10] and Zhou et al. [27] revealed that the land-air heat transfer in the Himalayas and Yarlung Zangbo River Valley in Nyingchi is strongly affected by the SASM. Due to the lack of observation data, the above studies mainly focused on the analysis of a single site or a typical underlying surface, but there was a lack of studies on the influence of the South Asian summer monsoon on the land-air exchange at other different sites in the Tibetan Plateau.

The network of plateau observation stations is sparse; the representativeness of observation stations is limited by the complex topography and underlying surface characteristics. The study of land-air interaction under the complex terrain of the plateau is much more difficult than that in other areas, the observation time, space, and physical-property variables are very limited. Because of this limitation, the third Tibetan Plateau Experiment for atmospheric science (TIPEX III) has been organized by China Meteorological Administration since June 2014, with nine boundary-layer observation stations established in the central, western, and southeastern parts of the plateau [28]. The observation sites are more widely distributed, and the data are the latest and most comprehensive, which provides an important data basis for the study of land-air energy exchange over the Tibetan Plateau. With this observational data, Wang et al. analyzed the surface parameters and near-surface turbulent fluxes over TP [29]. Most previous studies on the SASM impacts are from one or two in-situ stations in the south or Southeast Tibet, while the SASM impacts may be the largest. In this study, a total of eight stations covering different regions of TP were applied, aiming to study the different phases of SASM (active/break periods) impacts on land-atmosphere heat transfer over different plateau regions. In addition, previous studies suggested the great impacts of SASM on the local TP heating, as well as the regional climate over the Southeast Tibet and South Tibet; however, whether these impacts extend northwards was not clear until now. Thus, one of our purposes was to understand the SASM-affected area and extension. In this paper, data and methods are introduced in Section 2, and the SASM evolution (transition of active and break phases), and its possible influences on the land-air heat transfers over different plateau regions are presented in Section 3. The discussion and conclusions are given in Sections 4 and 5, respectively.

## 2. Data and Methods

The data used in this paper are from the third Tibetan Plateau (TP) Experiment for atmospheric science (TIPEX III) from late July to early September, 2014. During the experimental period, 9 observation stations were installed over the plateau regions. These stations are Ali, Nagqu, Amdo, Nyainrong, Biru, Baingoin, Lhari, Nyingchi, and Namco (see Figure 1 for the topography and Table 1 for the detailed station locations). At each station, the radiation fluxes (downward shortwave radiation flux and net radiation flux) were measured by a 4-component net radiometer (NR01, Hukseflux Thermal Sensors, Delftechpark, The Netherlands), and the land-atmosphere heat transfers (sensible and latent heat fluxes) were measured by a 3-D ultrasonic anemometer (CSAT3, Campbell Scientific, Inc., Logan, UT, USA). These raw data were calculated as the averaging interval of 30 min for analysis in this paper. In this paper, the total heat flux is defined as the sum of sensible and latent heat flux.

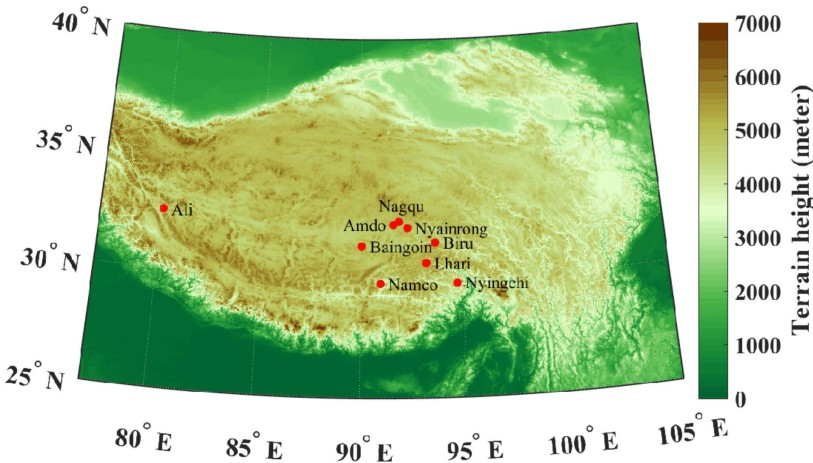

**Figure 1.** Topography of the Tibetan Plateau, with 9 plateau stations being denoted by red dots. It should be noted that the Nyingchi station was excluded from our study after the data quality control steps were completed.

**Table 1.** Station locations over the plateau regions.

| Stations | Location | Elevation (m) | Plateau Regions |
| --- | --- | --- | --- |
| Ali | 80.1°E, 32.5°N | 4350 | northwest plateau |
| Nagqu | 91.9°E, 32.4°N | 4509 | central plateau |
| Amdo | 91.6°E, 32.2°N | 4695 | central plateau |
| Nyainrong | 92.3°E, 32.1°N | 4730 | central plateau |
| Biru | 93.7°E, 31.5°N | 4408 | central plateau |
| Baingoin | 90.1°E, 31.4°N | 4700 | central plateau |
| Lhari | 93.2°E, 30.7°N | 4500 | central plateau |
| Namco | 91.0°E, 29.8°N | 4730 | southeast plateau |
| Nyingchi | 94.7°E, 29.8°N | 3327 | southeast plateau |

EDDYPRO (version 5.1) software (from Li-COR Corporation) is also used for turbulent flux data quality control [29–31]. After quality control steps were performed, Nyingchi station was excluded from our study due to the missing data of more than 30%. For the following analysis, 29 July–26 August was selected as the observational period when data were available from all 8 stations.

In addition to the observational data, the large-scale reanalysis data from ECMWF (European Centre for Medium-Range Weather Forecasts) Interim were also used, including wind and specific humidity, with a horizontal resolution of $0.75° \times 0.75°$. The interpolated outgoing long-wave radiation (OLR) data from NOAA (National Oceanic and Atmospheric

Administration) were applied to illustrate the convection conditions, with a horizontal resolution of $1.0° \times 1.0°$.

## 3. Results

### 3.1. SASM Evolution and Synoptic Situations

The SASM evolution could have great impacts on the weather and climate in Asia through general circulation changes [1,17,21,23,27,32]. The onset of SASM usually occurs at the end of May or early June in South Asia [33]. To investigate the impacts of the SASM on the land-air heat exchange processes over TP, the SASM evolution during the TIPEX III experiment in 2014 is first analyzed.

To characterize the SASM evolution in the TIPEX III in 2014, a SASM index (SASMI) from Wang et al. [34] is adopted in this study. The SASMI is defined by the standardized difference of averaged zonal wind speeds at 850 hPa from two regions, 5–15°N, 40–80°E, and 20–30°N, 70–90°E. A large positive SASMI corresponds to a strong monsoonal circulation, while a large negative SASMI corresponds to a weak monsoon. Figure 2 presents the daily variations in SASMI during the observation period from 1 May to 30 September 2014. This figure shows that the SASMI turns positive on 6 June, and then begins with a sudden increase to a maximum on 11 June 2014, with the maximum SASMI value being larger than the average value (5.0 m/s). At this time, a strong cyclonic circulation prevailed over the Arabian Sea at 850 hPa (figure not shown), which represents the SASM onset in 2014. Thereafter, SASM experienced several active periods during 11–16 July, 22 July, 29 July–1 August, 5 August, and 27 August–2 September, with positive SASMI exceeding the standard deviation, and break periods during 21–23 June, 28–30 June, 13–15 August, 24–25 August, with negative SASMI values exceeding the standard deviation. Considering the observation period, 29 July–1 August, and 5 August were selected as the SASM active period, and 13–15 August, and 24–25 August were selected as the SASM break period. In order to minimize the influence of the solar altitude angle, the active and break periods were selected close to each other. In the following studies, the land-air heat transfers, as well as radiation fluxes, will be averaged for the SASM active and break periods, to investigate their responses to the SASM evolution.

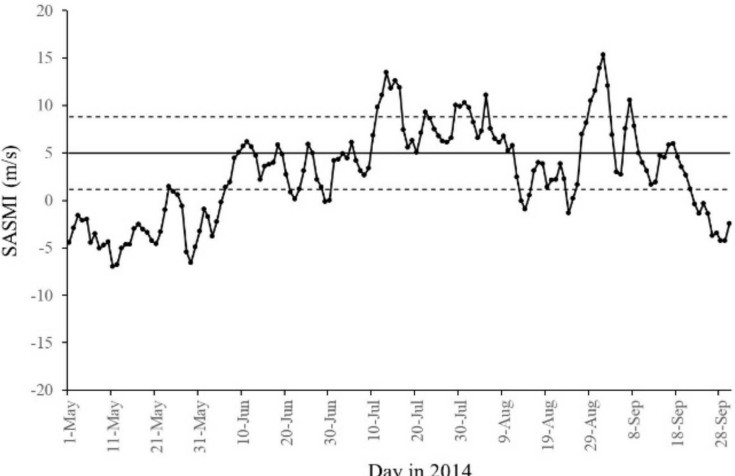

**Figure 2.** Variations in the SASM index (SASMI), with the averaged SASMI value (5.0 m/s) shown as a solid line, dashed lines represent one standard deviation (3.8 m/s) above and below the mean value.

To illustrate convection related to the SASM evolution, the outgoing long-wave radiation (OLR) was averaged for the entire observational period and for the SASM active and break periods; their distributions are shown in Figure 3. The low OLR values represent strong convection and vice versa. During the observational period (Figure 3a), there were three strong convective activity centers (with OLR values lower than 190 $Wm^{-2}$) in the eastern part of the Bay of Bengal, the northeastern part of India and the central portion of

TP. During the SASM active period (Figure 3b), the strong convective activities over the Bay of Bengal and North India both intensified and extended northward, with central OLR values less than 160 Wm$^{-2}$. The convection over the central TP became more severe and enlarged, with central OLR values lower than 160 Wm$^{-2}$ and covering almost the entire TP region. During the break period (Figure 3c), however, all the three convections moved southwards, and the convective activity center over the TP as shown in Figure 3a retreated to the south of TP. Therefore, obvious differences can be found in the OLR distributions between the SASM active and break periods, especially over the TP regions, representing the dominant strong and weak convections there.

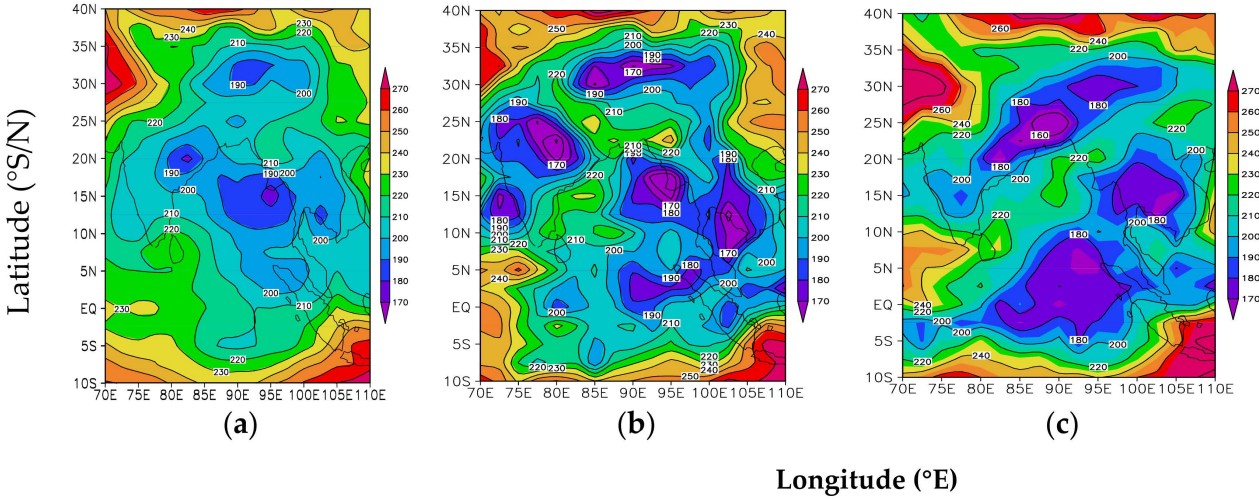

**Figure 3.** Outgoing long-wave radiation (OLR) distributions over the South Asia and TP regions, averaged for the (**a**) observation period, (**b**) SASM active period, and (**c**) SASM break period, with a contour interval of 10 Wm$^{-2}$.

Figure 4 presents the averaged wind and specific humidity fields at 500 hPa in the entire observation period, SASM active and break periods. During the observation period (Figure 4a), a cyclone formed over Central India and the Bay of Bengal, with the highest specific humidity greater than 4.5 g/kg. A westerly existed in the west TP, bringing dry air masses there (with the specific humidity less than 3.0 g/kg). A southwesterly prevailed over the south and central plateau regions, with specific humidity values greater than 5.5 g/kg. During the SASM active period (Figure 4b), the cyclone over Central India and the Bay of Bengal intensified, and moved northward and westward, associated with an enhancement of moisture (central specific humidity greater than 5.5 g/kg). An obvious cyclonic circulation appeared over the main body of the plateau. The dry westerly prevailed over the west TP weakened. The southwesterly existed in the south and Central TP became stronger, leading to higher moisture levels over the entire TP, with central values greater than 7.0 g/kg. During the SASM break period (Figure 4c), the cyclone with high water vapor over Central India and the Bay of Bengal as shown in Figure 4b disappeared. The dry westerly over the west and southwest of TP strengthened significantly, with moisture values below 2.0 g/kg. The weakened southwesterly led to the retreat and shrinking of the high moisture center over the south and Central TP.

### 3.2. The Impacts of SASM Evolution on the Radiation Heat Transfers

From the above results, large differences between the SASM active and break periods were observed from the synoptic situations, including the convection, wind, and moisture fields over the South Asian and TP regions. In the following study, the impacts of SASM evolution on land-air heat exchange processes, as well as the radiation conditions will be covered.

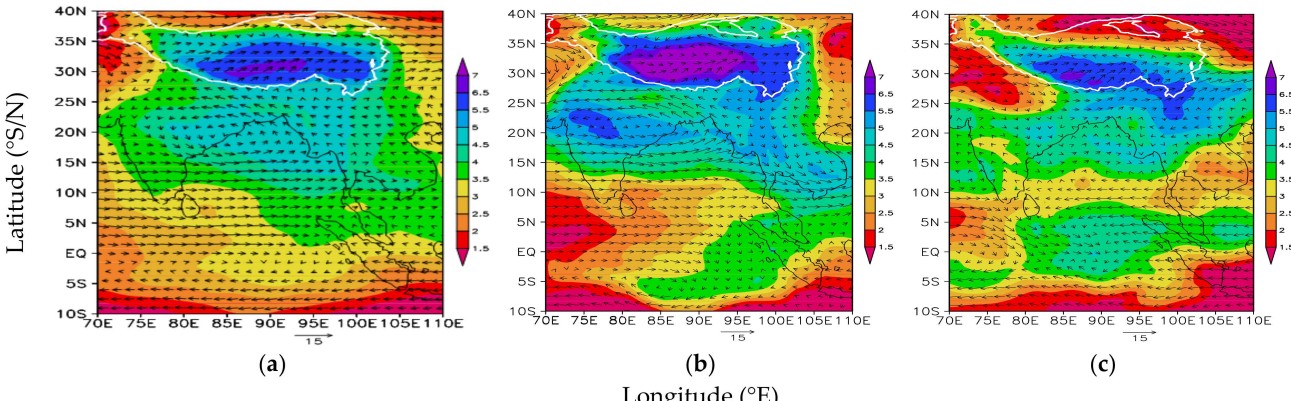

**Figure 4.** Horizontal distributions of wind (arrows, units: m/s) and specific humidity. (shadings, units: g/kg) at 500 hPa averaged for (**a**) the observation period, (**b**) the SASM active period, and (**c**) the SASM break period.

Figure 5 shows the diurnal variations of the downward shortwave radiation flux (DR), averaged for the observations period, SASM active, and break periods. The diurnal variations of DR are similar for all stations, showing increases at approximately 06:00 LST (local standard time) near sunrise, reaching a maximum at noon, and decreasing to almost zero at approximately 18:00 LST near sunset. However, large DR amplitude differences were found among the 8 plateau stations. The strongest DR occurred over Ali station (northwest plateau), with daily average and maximum values of 319.2 and 1007.8 Wm$^{-2}$, respectively, which is mainly due to the low precipitation and less moisture in the air. Ali station is located in the northwest of the Qinghai-Tibet Plateau, and Ali is a very dry area with very little rainfall. According to the observation data, Ali station had no precipitation from July to September in 2014, resulting in low water vapor content in the air. The second strong DR is found over Namco station (southeast plateau), with daily averaged and maximum values of 222.9 and 846.3 Wm$^{-2}$, respectively. For the other stations, the DR differences are noticeably smaller, and the difference of the daily averaged values was less than 20 Wm$^{-2}$, with a variation between 187.3 Wm$^{-2}$ and 206.1 Wm$^{-2}$. During the SASM active/break periods, the DR is greatly weakened/strengthened at most plateau stations. For example, the daily averaged value of DR at the Baingoin station was 194.2 Wm$^{-2}$ during the observation period, and the DR varies from 133.2 Wm$^{-2}$ (weakened by 31.4%) during the SASM active period and 234.5 Wm$^{-2}$ (strengthened by 20.8%) during the break period, respectively. The weakened/strengthened DR is closely related to the strong/weak convections over the plateau region during the SASM active/break periods (see Figure 3). The strong/weak convections can result in more/less cloudiness, which further affects the solar radiation by blocking/enhancing effects [17,27,32].

Figure 6 presents the net radiation fluxes (NR) for the 8 stations over the TP during the observations, SASM active, and break periods. The NR patterns exhibit similar diurnal variations as those of the DR, with positive values (net heating) during daytime from approximately 06:00 LST to approximately 18:00 LST and negative values (net cooling) for the other times of the day over most of the plateau stations. NR amplitude differences are also seen among the 8 plateau stations. The strongest net radiation also occurs at Namco station (southeast plateau), with daily averaged and maximum values of 133.2 and 609.2 Wm$^{-2}$, respectively. The second strongest NR is found over Lhari station (central plateau), with a diurnally averaged value of 120.0 Wm$^{-2}$ and a maximum of 511.6 Wm$^{-2}$. Over the other stations, the NR differences are quite small; the difference in the daily averaged value was about 10 Wm$^{-2}$, with a range from 102.8 to 112.2 Wm$^{-2}$. The NR is also weakened/strengthened during the SASM active/break periods at most of the stations compared with the observational mean. As with DR, the largest effects of SASM on the NR occurred at Baingoin station, the daily averaged value of NR was 102.8 Wm$^{-2}$ during the observation period, while the NR was 66.9 Wm$^{-2}$ during the SASM active period, which

weakened by 34.9%, and the NR was 128.8 Wm$^{-2}$ during the SASM break period, which strengthened by 25.3%.

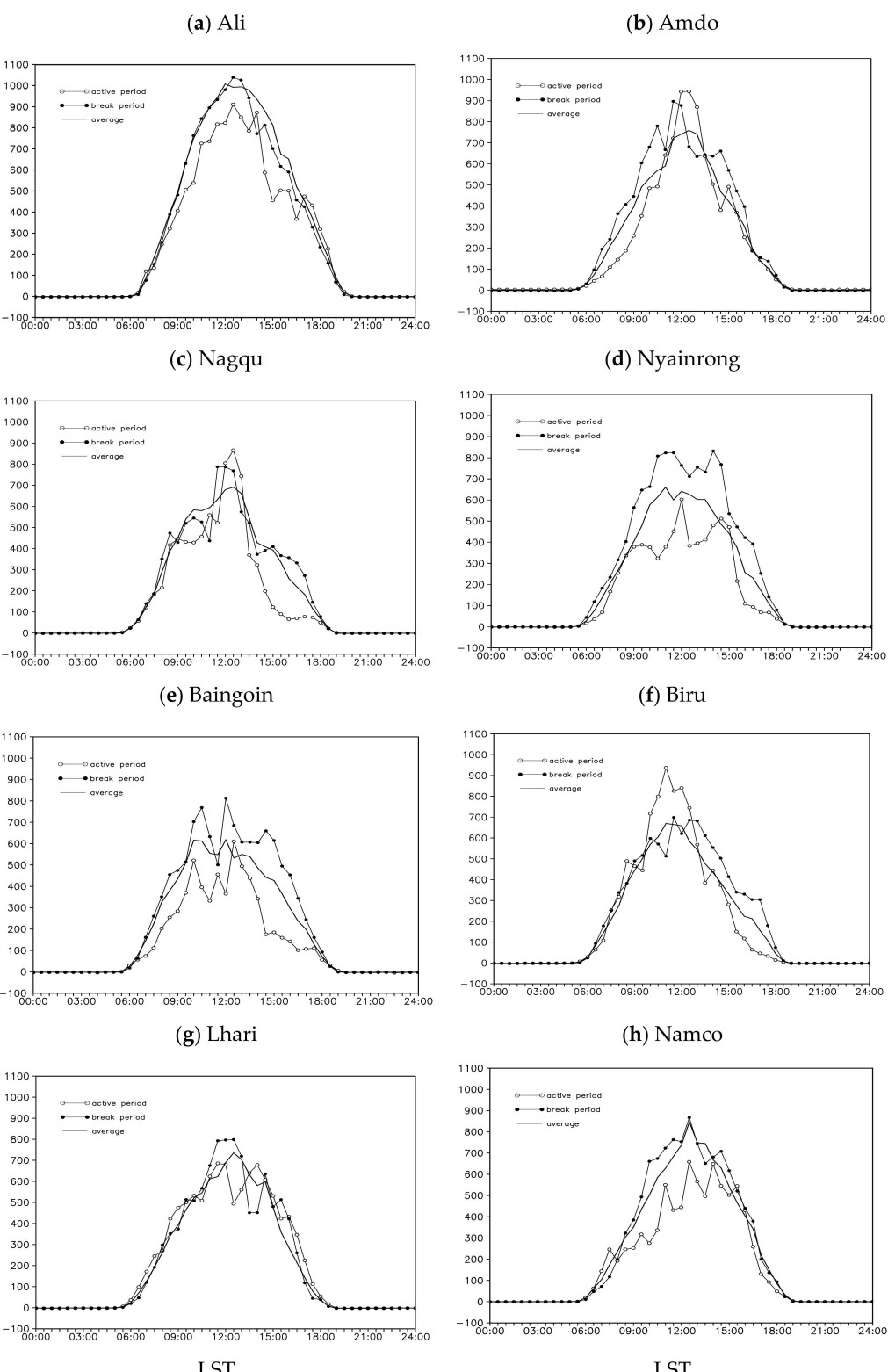

**Figure 5.** Diurnal variation of the downward short-wave radiation flux (DR) (units: Wm$^{-2}$) from 8 stations, averaged for the observations, SASM active, and break periods.

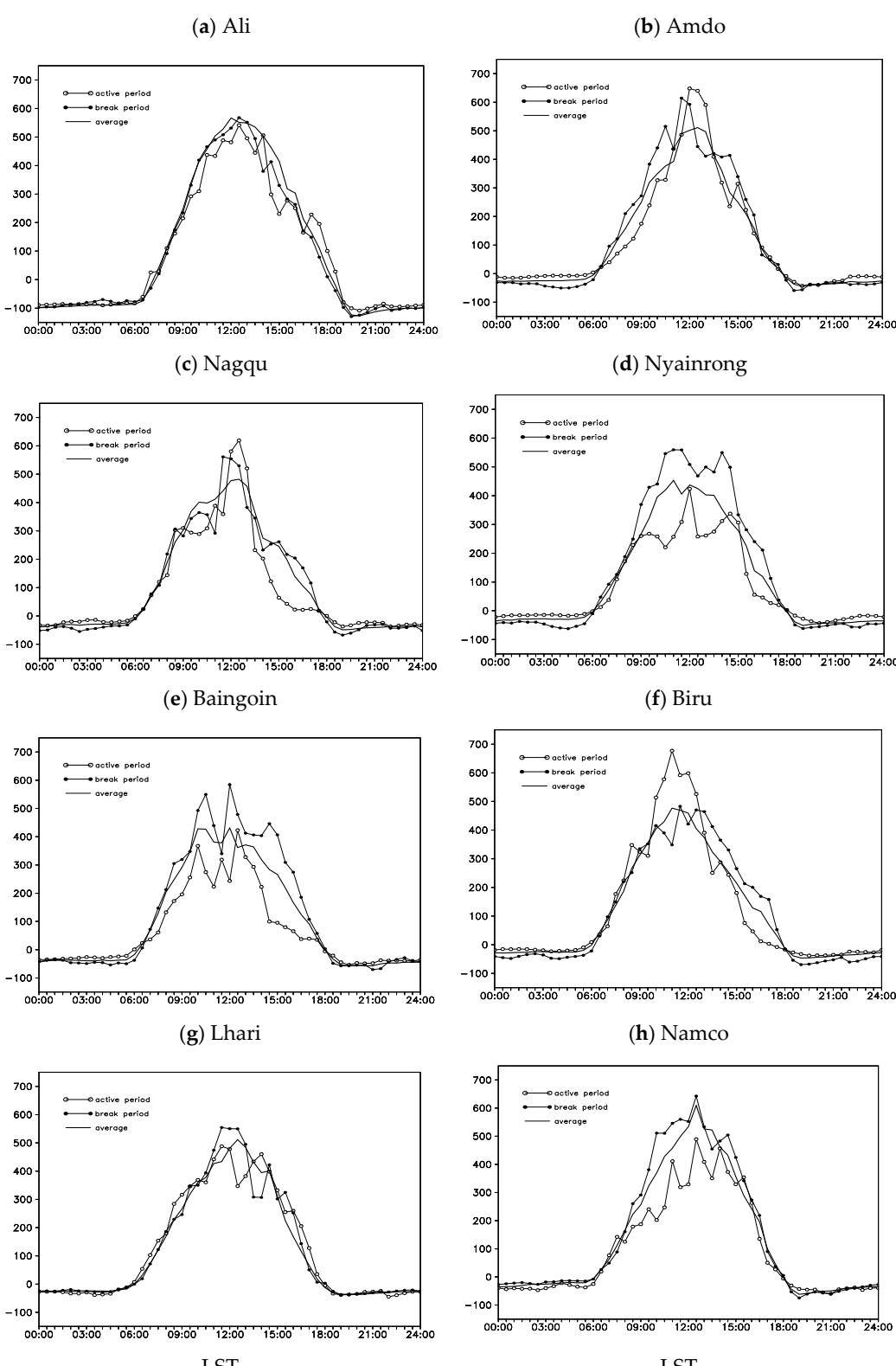

**Figure 6.** Same as for Figure 5, but for the net radiation flux (NR) (units: Wm$^{-2}$).

### 3.3. The Impacts of SASM Evolution on the Turbulent Heat Transfers

Figure 7 shows the diurnal variations in sensible heat flux (SH), averaged for the observations, SASM active, and break periods. Driven by the net heating (see Figure 6), the plateau releases heat into the atmosphere during the daytime (positive SH values) and receives heat from the atmosphere at night (negative SH values). Following the diurnal

variation in radiation, the sensible heat flux increases from early morning at approximately 6:00 LST, reaches a maximum at noon, and decreases later in the day. Differences are clearly found in the SH averaged and maximum values among the 8 plateau stations, despite the similar diurnal variations. During the observation period, the largest sensible heat transfer occurs over Ali station (northwest plateau), with daily averaged and maximum values of 60.1 (see Table 2) and 197.4 $Wm^{-2}$, respectively. The second-largest SH is found over Namco station (southeast plateau), with daily averaged and maximum values of 28.2 and 118.9 $Wm^{-2}$, respectively. Over the other stations, the diurnally averaged SH varies from 20.0 to 26.5 $Wm^{-2}$, and the daily maximum values vary from 73.1 to 98.2 $Wm^{-2}$. The smallest SH occurs over the Biru station (central plateau), with averaged and maximum values of 18.8 and 77.5 $Wm^{-2}$, respectively. Our results are consistent with recent study results. The recent research reveals that SH in the central TP in August is generally between 5 and 40 $Wm^{-2}$, with an average of 18 $Wm^{-2}$, while SH in the western TP is between 40 and 70 $Wm^{-2}$, with an average of 56 $Wm^{-2}$, these results are also smaller than that in the past [29,35]. Ye and Gao [1] estimated the July-August-mean intensity of SH is 60–80 $Wm^{-2}$ over the central TP and 150–190 $Wm^{-2}$ in the western TP, and Yang and Guo [36] estimated SH in July-August is 50–60 $Wm^{-2}$ in the central TP and 75–90 $Wm^{-2}$ in the western TP, remarkably larger compared to the new findings. This result indicates that SH has been possibly overestimated by the previous studies when calculating SH using the bulk transfer method, which is based on the larger values of the bulk transfer coefficient for sensible heat [28].

During the SASM active period, the response of land-to-atmosphere sensible heat transfer exhibited great differences among the 8 stations. For example, the daily averaged SH weakened at most stations, with the amplitude of weakening varying from −1.7% to −41.4%. The largest weakening occurred at Namco station (southeast plateau), and the SH was 16.5 $Wm^{-2}$ over the SASM active period, which was weakened by 41.4% from its daily averaged value of 28.2 $Wm^{-2}$ during the observation period. The smallest weakening happened at Biru station (central plateau), with a weakening of 1.7% from its daily averaged value. At stations Amdo and Nyainrong, the daily averaged SH values strengthened, with the amplitudes increasing by 12.3% and 13.2%, respectively. During the SASM break period, the SH largely strengthens over all stations, and with the strengths varying from 0.8% (at Ali) to 45.3% (at Nyainrong). Therefore, the sensible heat transfers over the plateau region can be affected by the SASM evolution, with the weakened/strengthened amplitudes over most stations during the SASM active/break periods, which are closely related to the weakened/strengthened radiation conditions [9,10,17,23,27,32]. However, these SASM impacts on the sensible heat transfer exhibit large inhomogeneity over the plateau regions. Overall, the more southerly stations received more SASM impacts. The larger SASM impacts on sensible heat transfers occurred at stations Namco, Baingoin, and Lhari, and the SH differences between the SASM active and break periods were greater than 38% of the daily averaged values. The SASM impacts also extended westward and northward to the Ali and Nagqu stations, with the SH differences between the SASM active and break periods reaching 21.4% and 34.6% of their daily averaged values, respectively. However, the SASM impact appeared to be negligible at Amdo station (south of Nagqu station), which complicated our results.

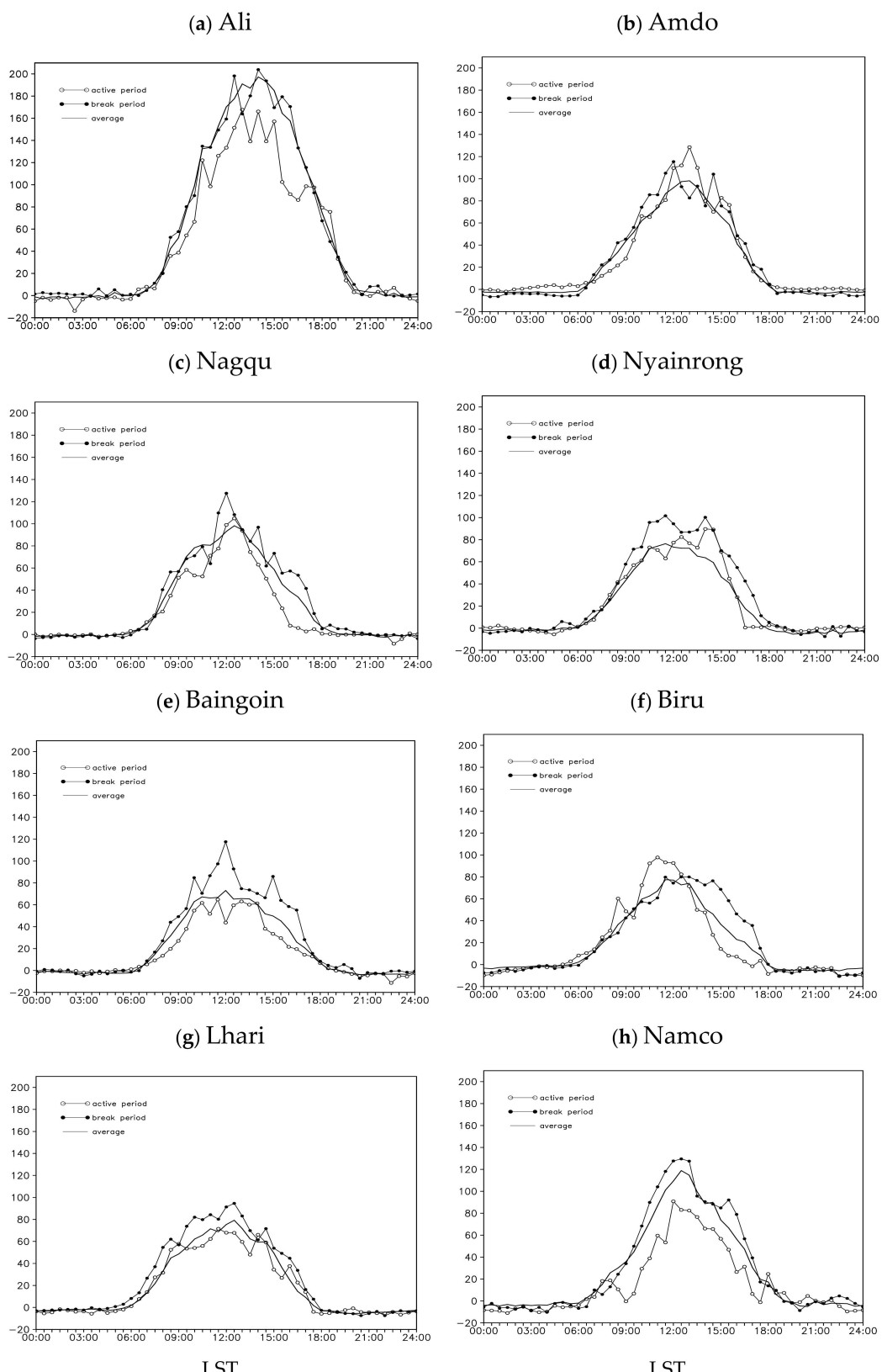

**Figure 7.** Diurnal variations of the sensible heat flux (SH) (units: Wm$^{-2}$) from 8 stations, averaged for the observations and SASM active and break periods.

**Table 2.** Sensible heat flux (SH) (Wm$^{-2}$) over the 8 plateau stations, averaged for the observations and SASM active and break periods. The bracketed values denote the SH percentage increase (decrease) of the daily averaged value, in which positive (negative) values mean increasing (decreasing).

| SH/Stations | Ali | Nagqu | Amdo | Nyainrong | Biru | Baingoin | Lhari | Namco |
|---|---|---|---|---|---|---|---|---|
| Observation | 60.1 | 26.5 | 24.5 | 20.0 | 18.8 | 20.2 | 20.0 | 28.2 |
| SASM active | 47.7 (−20.5%) | 20.9 (−21.2%) | 27.5 (12.3%) | 22.6 (13.2%) | 18.5 (−1.7%) | 16.0 (−20.8%) | 18.6 (−6.8%) | 16.5 (−41.4%) |
| SASM break | 60.6 (0.8%) | 30.1 (13.4%) | 27.0 (10.1%) | 29.0 (45.3%) | 21.7 (15.4%) | 27.7 (37.6%) | 26.3 (31.7%) | 30.9 (9.7%) |

Figure 8 presents the diurnal variation of latent heat flux (LH), averaged for the observations, SASM active, and break periods. Differing from the sensible heat flux, the latent heat over the TP is always transferred upwards (positive LH values) during the entire day. In addition, the amplitude of LH was much larger than that of SH over most of the plateau stations and that was consistent with previous results [37]. Obvious differences are seen in the LH daily averaged and maximum values among the 8 plateau stations. During the observation period, the largest latent heat transfer occurred at Nagqu station (central plateau), with daily averaged and maximum values of 74.7 (see Table 3) and 238.6 Wm$^{-2}$, respectively. The smallest LH occurred at Ali station (northwest plateau) due to small amounts of precipitation there, with daily averaged and maximum values of 10.1 and 27.2 Wm$^{-2}$, respectively. Over the other stations, the diurnally averaged LH varied from 53.0 to 74.4 Wm$^{-2}$, and the maximum values varied from 152.8 to 245.7 Wm$^{-2}$.

**Table 3.** Latent heat flux (LH) (Wm$^{-2}$) over the 8 plateau stations, averaged for the observations and SASM active and break periods. The bracketed values denote the LH increasing (decreasing) percentage of the daily averaged value, in which the positive (negative) values mean increasing (decreasing).

| LH/Stations | Ali | Nagqu | Amdo | Nyainrong | Biru | Baingoin | Lhari | Namco |
|---|---|---|---|---|---|---|---|---|
| Observation | 10.1 | 74.7 | 74.4 | 66.6 | 59.3 | 55.4 | 53.0 | 65.0 |
| SASM active | 18.9 (87.4%) | 56.8 (−24.0%) | 78.7 (5.9%) | 72.8 (9.4%) | 69.5 (17.3%) | 47.5 (−14.3%) | 48.3 (−8.8%) | 59.1 (−9.0%) |
| SASM break | 5.9 (−41.2%) | 77.5 (3.8%) | 73.5 (−1.1%) | 77.6 (16.6%) | 63.5 (7.1%) | 62.9 (13.6%) | 57.8 (9.0%) | 63.0 (3.0%) |

Compared with the impacts of the SH, the SASM impacts on LH were relatively small and complicated. At stations Namco, Lhari, and Baingoin, while the SASM impacts on the SH were large, but the SASM impacts on LH were smaller, with the LH differences between the SASM active and break periods varying from 12.0% to 27.9% of their daily averaged values. The small impacts over these stations could be closely related to the high moisture conditions there. The SASM impacts could also extend to the north plateau, and Nagqu station with a large LH difference (27.8% of the daily averaged value) between the SASM active and break periods. However, the same as for SH, the SASM impacts on LH seem negligible over station Amdo. It should be noted that the SASM impact over Ali could be ignored due to the small LH value despite having the largest LH response amplitude of the daily averaged values.

The total heat transfer (TH) is defined as the sum of SH and LH. Figure 9 shows the diurnal variation in TH, averaged for the observations, SASM active, and break periods. Clear differences can be seen in the TH magnitudes among the 8 plateau stations. During the observation period, the largest TH occurred at Nagqu station (central plateau), with daily averaged and maximum values of 101.2 (see Table 4) and 336.7 Wm$^{-2}$, respectively. The smallest TH occurred over Ali station (northwest plateau), with daily averaged and maximum values of 70.2 and 220.3 Wm$^{-2}$, respectively. Over the other stations, the

diurnally averaged TH varied from 73.0 to 98.8 Wm$^{-2}$, and the daily maximum values varied from 225.9 to 343.9 Wm$^{-2}$.

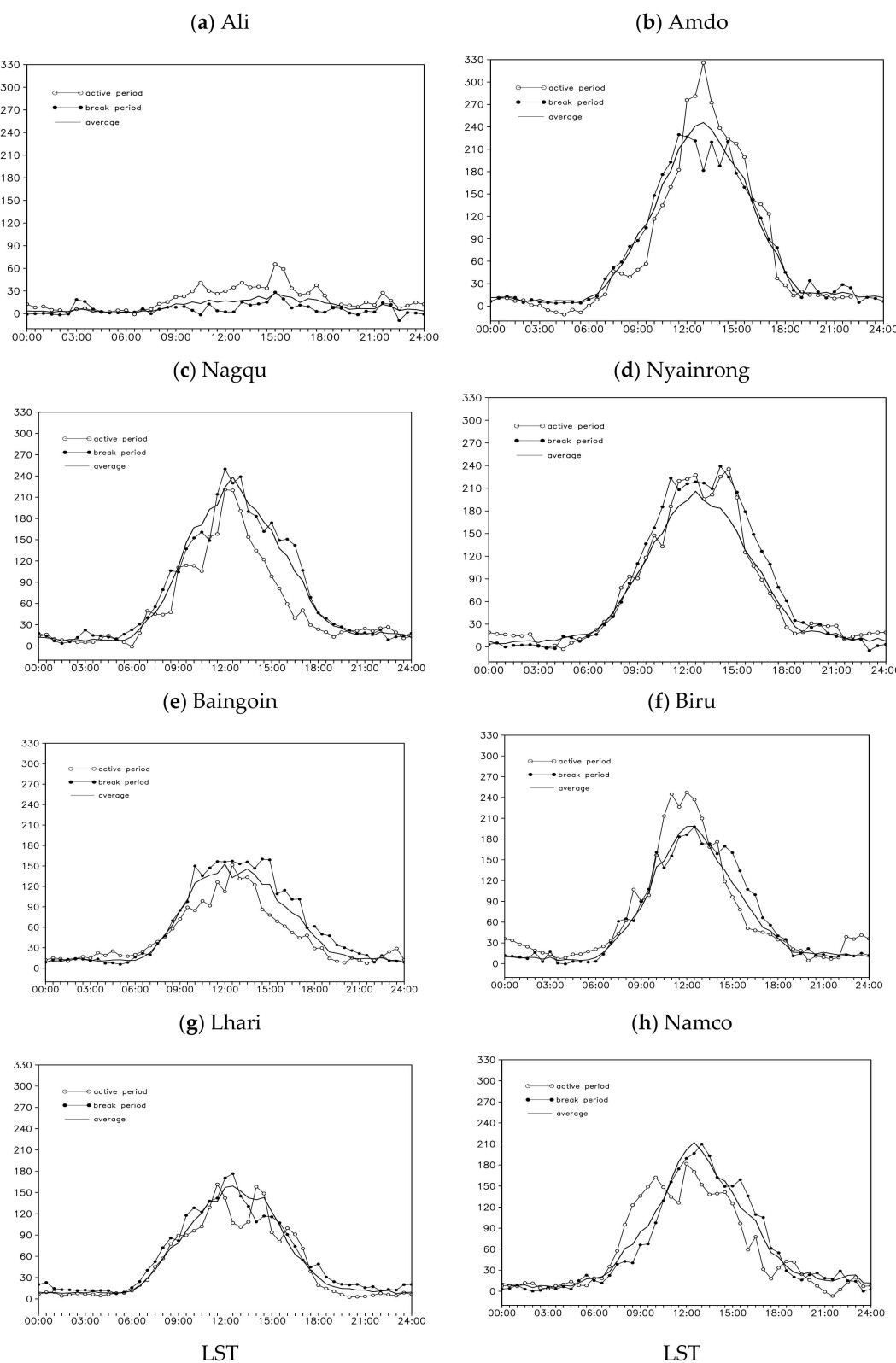

**Figure 8.** Same as Figure 7, but for latent heat flux (LH) (units: Wm$^{-2}$).

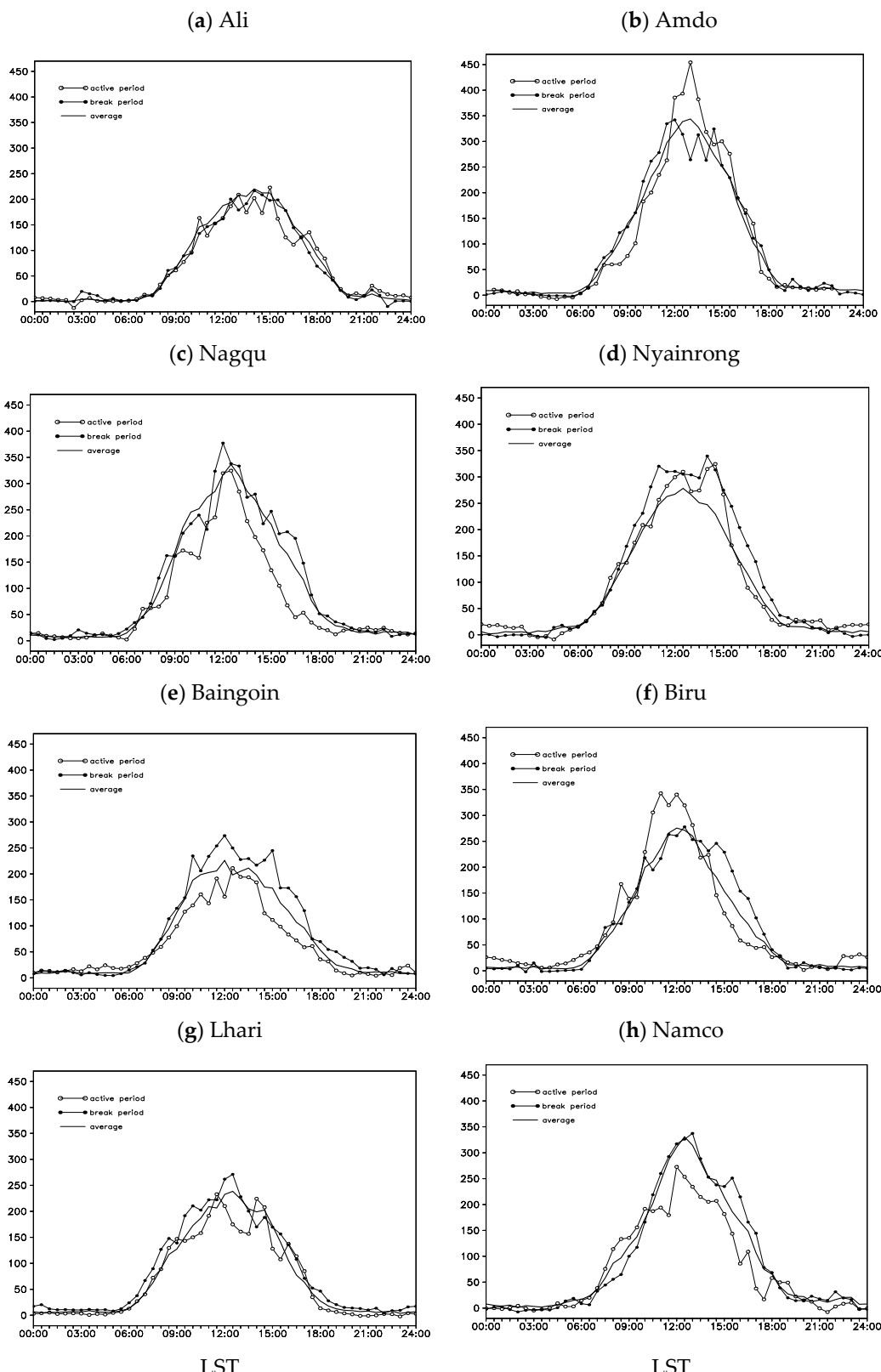

**Figure 9.** Same as Figure 7, but for total heat flux (TH) (units: Wm$^{-2}$).

**Table 4.** Total heat flux (TH) ($Wm^{-2}$) over the 8 plateau stations, averaged for the observations and SASM active and break periods. The bracketed values denote the TH increasing (decreasing) percentage of the daily averaged value, in which the positive (negative) values mean increasing (decreasing).

| TH/Stations | Ali | Nagqu | Amdo | Nyainrong | Biru | Baingoin | Lhari | Namco |
|---|---|---|---|---|---|---|---|---|
| Observation | 70.2 | 101.2 | 98.8 | 86.5 | 78.1 | 75.5 | 73.0 | 93.1 |
| SASM active | 66.7 (−5.0%) | 77.7 (−23.3%) | 108.7 (10.0%) | 95.4 (10.2%) | 88.0 (12.7%) | 63.4 (−16.0%) | 67.0 (−8.3%) | 75.6 (−18.8%) |
| SASM break | 66.5 (−5.2%) | 107.6 (6.3%) | 100.5 (1.7%) | 106.6 (23.2%) | 85.2 (9.1%) | 90.6 (20.0%) | 84.1 (15.2%) | 93.9 (0.8%) |

From Table 4, the SASM impacts on the TH can be clearly seen at stations Namco, Baingoin, Lhari, and Nagqu, with a weakened/strengthened magnitude during the SASM active/break period, and the TH differences in daily averaged value between the SASM active and break periods are with a range between 19.6% and 36.6%. The largest impacts occurred at Baingoin station, with the TH difference reaching 36.0% of the daily averaged value between the SASM active and break periods. In comparison, the SASM impacts on TH over the other plateau stations were quite small or even negligible.

## 4. Discussion

In theory, there are multi-scale atmospheric motions, especially over the complex TP regions, which interact with each other. The local land-atmosphere heat transfers over TP could be possibly affected by the large-scale circulations, such as the surrounding South Asia monsoon systems. Theoretically, the monsoon could affect the local heat transfer by adjusting the local radiation conditions through cloud and precipitation variations, which have been proved by previous studies [9,23,27].

From our results, not all stations show consistent impacts from SASM. For example, the Biru station has an opposite variation of heat flux during the different SASM stages, compared with other stations at similar latitudes, such as Baingoin and Lhari. The reason is still unclear; this may have connections with orographic peculiarities and microclimatic features, and which could be partly leading to the different local heating forces there. The problem also needs to be further studied using longer-time observation data in the future.

The spatial differences between various sites could be possibly not only affected by the SASM activities. Our study presented a clue that the local land-air exchange processes could be strongly affected by large-scale circulations, such as the SASM evolution, with inhomogeneous distributions over the whole TP region. The local topographic effects should be carefully considered for future studies.

Due to the hard living and working conditions, the observational data are quite limited over Tibet, especially over the northern and western parts of the plateau. Thus, as shown in our introduction, most previous studies focused mainly on the SASM impacts over the south or the southeast Tibet, one reason is due to the largest impacts there which were shown from our studies, and another reason is no observational data over the other plateau regions. During the most recent year, the Third Tibetan Plateau Atmospheric Scientific Experiment provided us with an opportunity to make a comprehensive study on the possible SASM impacts on different regions of TP. Although our conclusions are based on only one year of data (2014), the physical mechanism is the same for the other year, in spite of year-to-year variations. The SASM evolution could result in cloud and advection variations, adjusting the local radiation conditions, and further affecting the local heat transfers. Therefore, the inhomogeneities of SASM impacts should be confirmed for each year, but varied with different amplitudes and monsoon extended regions. In the future, based on more accumulated data, further investigations and more evidence are still needed for the robustness of conclusions.

## 5. Conclusions

Using the observation data from TIBEX III, as well as the large-scale reanalysis data from ECMWF Interim and from NOAA, the different impacts of SASM on the land-atmosphere heat exchange processes between active and break periods over different regions of TP were investigated. During the observation period (29 July to 26 August), the land-atmosphere heat transfer exhibited strong inhomogeneous distributions over the plateau regions. The daily average total heat transfer varied from 70.2 to 101.2 Wm$^{-2}$ over the 8 plateau stations, with the sensible heat flux ranging from 18.8 to 60.1 Wm$^{-2}$ and the latent heat flux with a variation between 10.1 and 74.7 Wm$^{-2}$. The latent heat transfer values larger than the sensible heat transfer values over most of the plateau regions are mainly related to the strong convection that prevailed over the plateau during the observation period (Figure 3a), which caused the high moisture conditions (Figure 4a). The land-atmosphere heat transfer can be largely affected by the SASM evolution, but with strong inhomogeneity over the plateau stations. Overall, the more southerly stations received more SASM impacts. The land-atmosphere heat transfers (the total, sensible, and latent heat fluxes) are greatly weakened/strengthened during the SASM active/break period at the Namco (southeast plateau), Baingoin (central plateau), Lhari (central plateau), and Nagqu (central plateau) stations, with the sensible heat flux differences between the SASM active and break periods varying from 34.6% to 58.4% of the daily averaged values, with a range between 12.0% and 27.9% for the latent heat flux and with a variation between 19.6% and 36.0% for the total heat transfer. These significant SASM influences could be closely related to the weakened/strengthened radiation conditions [9,10,17,23,27,32]. However, the impacts of SASM during active/break periods become complicated over the other plateau stations. For example, the impact of SASM on Ali station is mainly reflected in the influence of sensible heat flux, because sensible heat flux plays a dominant role in the total heat transfer. The different phases of SASM impacts on the Amdo and Nyainrong stations were quite small or even negligible, which complicates our conclusions. Therefore, further investigations are still needed based on more observational data over a long period in the Tibetan Plateau.

**Author Contributions:** H.L.: Formal analysis, Writing—original draft. L.Z.: Conceptualization, Supervision, Writing—review & editing. G.W.: Formal analysis, Writing—review & editing. All authors have read and agreed to the published version of the manuscript.

**Funding:** This research was funded by the Strategic Priority Research Program of the Chinese Academy of Sciences (Grant No. XDA19070401), the National Natural Science Foundation of China (Grant 42030611), the Second Tibetan Plateau Scientific Expedition and Research (STEP) program (grant No.2019QZKK0103 and 2019QZKK0105), the National Natural Science Foundation of China (Grant 91937301 and 41830968), the CAS Key Subordinate Project (KGFZD-135-16-023), the Forecaster Special Project of China Meteorological Administration (No.CMAYBY2019-155), the Heavy Rain and Drought-Flood Disasters in Plateau and Basin Key Laboratory of Sichuan Province (SC-QXKJYJXMS202116), and the Opening Foundation of Plateau Atmosphere and Environment Key Laboratory of Sichuan Province (grant No.PAEKL-2020-C7).

**Data Availability Statement:** The data that support the findings of this study are available from the first author upon request (Hongyi Li, lihongyi@cma.gov.cn).

**Acknowledgments:** We appreciate the access to the ECMWF Interim and NOAA datasets. The authors appreciate all of the hard work done by researchers attending the third Tibetan Plateau (TP) Experiment (TIPEX III).

**Conflicts of Interest:** The authors declare that they have no known competing financial interests or personal relationships that could have appeared to influence the work reported in this paper.

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
