# Peer review of "The Observed Impact of the South Asian Summer Monsoon on Land-Atmosphere Heat Transfers and Its Inhomogeneity over the Tibetan Plateau"

_remotesensing, doi:10.3390/rs14133236_

Round 1
Reviewer 1 Report
The authors present a useful data paper in which they share observations of turbulent heat flux from flux towers deployed in the Tibetan Plateau. The primary purpose of the study is to present these data and to describe how fluxes vary between active and break phases of the South Asian monsoon. The authors provide some discussion of the possible reasons for geographic variability between stations, leaving some questions open for future investigation.
Data papers like this are a valuable service to the research community, and this particular paper is very clearly presented. I expect that it can be published with minimal revision. My one major question--for the authors and for the editors--is whether Remote Sensing is the best MDPI outlet for the study. In a formal sense, anemometers and tower radiometers are "remote sensing," but I'm used to seeing airborne and spaceborne remote sensing papers in this journal. I might think that a journal focused on climate or hydrology would be a better outlet for this paper. But that's up to the authors and editor.
My only (minor) specific comment is that in a number of passages the authors refer to "SASM impacts" on fluxes, when they are in fact referring to "impacts of SASM break/active variability" on fluxes. On my first read I was confused and thought that "SASM impacts" referred to seasonal or interannual SASM impacts, so I would recommend that the authors phrase more precisely in these passages to emphasize the subseasonal or active/break meaning of the phrase.
Typo: in line 60, the sentence should read "...(SASM) is an important..."
Reviewer 2 Report
Dear Authors:
I believe that the analysis of any direct experimental data on the surface heat fluxes especially in such specific regions as Tibetan Plateau is interesting and important. However, your manuscript rather looks like technical report than as scientific paper. You write that the South Asian summer monsoon impacts significantly the land-air heat transfer at the some observational site, but this impact is small or even negligible in the other ones. However, the relevant analysis of this principal result is absent. Instead of that your describe and discuss in detail the different components of the total heat fluxes using numerous tables. It seems to me you need to analysis in detail at least two following things.
1. You consider two episodes of the active SASM phase and two episodes of the SASM break period. Then you analyze the heat fluxes' difference between these two SASM phases. But what about differences between two episodes within one SASM phase?
2. If the spatial difference between various sites is significant further analysis is needed. For instance, you can consider the orographic peculiarities in the vicinity of different sites or/and microclimatic features. The goal of such analysis is to explain the difference between various sites. I agree with you that further investigations are still needed, but I think they are needed in the discussed manuscript.
Reviewer 3 Report
This study with a title of “The Observed Impact of the South Asian Summer Monsoon on Land-atmosphere Heat Transfers and its Inhomogeneity over the Tibetan Plateau” would like to investigate the impact of the South Asian Summer Monsoon on land-atmosphere heat transfers over the Tibetan Plateau, which is of importance for better understanding the heating role of this plateau and thus the climate system. However, I find that there are some gaps in this study, which could be found below. So, I would like to recommend it to be conducted a major revision before considering publication.
1. In this study, I have a great confusing issue, i.e., what is your reasons that the SASM can impact the land-atmosphere heat transfers? Please show the theoretical evidences.
2. Maybe, it is not reasonable to directly various heat fluxes during SASM active and break periods, because the two periods have different solar altitude angles (in spite of the days between the two periods being close to each other), implying that the differences in shortwave radiation exist. So, what magnitudes the differences in the solar altitude angles during the two periods could impact you results?
3. When you compared various heat fluxes at a given site during different periods, how did you conclude the impacts of local meteorological conditions or local land-air interactions (i.e., who results in differences in cloudiness?) on your results? In other word, how did you know the differences caused by the SASM evolution rather than the local meteorological conditions or local land-air interactions?
4. When you compared the heat fluxes among different sites, it is better to consider the differences in locations of these sites (e.g., different latitude correspond differences in background of solar shortwave radiation), and its surrounding environments (e.g., aspect, and slope). These may introduce some uncertainties into your findings.
5. There are some confusing expressions in this manuscript, such as “with a positive standard deviation of several days” (L134). Please carefully check this version.
Round 2
Reviewer 2 Report
I believe that the manuscript can be published after minor text editing (see below) because any experimental data on the surface heat fluxes, especially in such specific region as Tibetan Plateau is interesting for the wide scientific community. English is not my native language but I see some inaccuracies, especially in the use of articles. For example, фе the line 447 one can read: ...the physical mechanism is same for the other year,... (should be "is the same...")
